# Negative Energy Balance Induced by Exercise or Diet: Effects on Visceral Adipose Tissue and Liver Fat

**DOI:** 10.3390/nu12040891

**Published:** 2020-03-25

**Authors:** Robert Ross, Simrat Soni, Sarah Houle

**Affiliations:** School of Kinesiology and Health Studies, Department of Endocrinology and Metabolism, Queen’s University, Kingston, ON K7L 3N6, Canada; s.soni@queensu.ca (S.S.); sarah.houle@queensu.ca (S.H.)

**Keywords:** exercise, abdominal obesity, energy balance, caloric restriction, non-alcoholic fatty liver disease, physical activity

## Abstract

The indisputable association between visceral adipose tissue (VAT) and cardiometabolic risk makes it a primary target for lifestyle-based strategies designed to prevent or manage health risk. Substantive evidence also confirms that liver fat (LF) is positively associated with increased health risk and that reduction is associated with an improved metabolic profile. The independent associations between reductions in VAT, LF, and cardiometabolic risk is less clear. In this narrative review, we summarize the evidence indicating whether a negative energy balance induced by either an increase in energy expenditure (aerobic exercise) or a decrease in energy intake (hypocaloric diet) are effective strategies for reducing both VAT and LF. Consideration will be given to whether a dose-response relationship exists between the negative energy balance induced by exercise or diet and reduction in either VAT or LF. We conclude with recommendations that will help fill gaps in knowledge with respect to lifestyle-based strategies designed to reduce VAT and LF.

## 1. Introduction

Decades of evidence have firmly established that VAT is associated with cardiometabolic risk factors beyond obesity per se [1]. There is no dispute that VAT represents a primary treatment target for strategies designed to prevent or manage the health risks associated with abdominal obesity. The findings from systematic reviews confirm that a negative energy balance induced by exercise or diet is associated with significant reductions in VAT and associated cardiometabolic risk factors [2,3]. Routine exercise consistent with current guidelines, in combination with a balanced diet, is associated with significant reductions in VAT. Whether exercise or a hypocaloric diet is associated with VAT reduction in a dose-response manner remains unclear. It is also unclear whether exercise intensity is positively associated with VAT reduction. What is clear, however, is that exercise or diet is associated with a reduction in VAT and associated cardiometabolic risk factors with minimal or no weight loss [4]. This is good news for practitioners who seek treatment options for their patients/clients.

Non-alcoholic fatty liver disease (NAFLD), an umbrella term that encompasses the deposition of lipid in the liver to more progressive steatosis and associated hepatitis, fibrosis, and cirrhosis, is recognized as an ectopic site of fat deposition that has serious health consequences [5]. Although LF accumulation is positively associated with VAT, there is evidence to suggest that LF is associated with cardiometabolic risk factors independent of VAT [6]. Thus, LF represents another target for strategies designed to reduce obesity-related health risk. In adults with or without NAFLD, exercise induced-negative energy balance consistent with current guidelines is associated with a marked reduction in LF.

In this narrative review, we summarize the evidence indicating whether a negative energy balance induced by either an increase in energy expenditure (aerobic exercise), or a decrease in energy intake (hypocaloric diet), are effective strategies for reducing both VAT and LF. The primary focus of this review was to report on findings from randomized trials with supervised aerobic exercise and monitoredhypocaloric diet interventions. Consideration was given to whether a dose-response relationship exists between the negative energy balance induced by exercise or diet and reduction in either VAT or LF. We conclude with recommendations that will help fill gaps in knowledge with respect to lifestyle-based strategies designed to reduce VAT and LF.

## 2. Visceral Adipose Tissue (VAT)

### 2.1. Measurement of VAT

Several methodologies for assessing VAT volume and distribution have been established. The criterion methods, which provide direct measurement of partial or total volumes, include magnetic resonance imaging (MRI) and computed tomography (CT) [4]. More recently, dual energy X-ray absorptiometry (DXA) has been identified as a reference method for assessing body composition [7]. However, due to high cost and feasibility issues, indirect estimates of VAT are often performed using anthropometric measures, including waist circumference, sagittal diameter, and bioelectrical impedance analysis [4]. When using MRI or CT to determine the effects of exercise or diet on VAT, it is important to note that change in VAT determined using a single image (e.g., a single image obtained at the level of the L4-L5 intervertebral space) is not materially different from the corresponding reduction in VAT derived using a multiple-image protocol [8]. Thus, when accessible, a single MRI or CT image can be used to accurately determine VAT distribution and/or reduction throughout the abdomen.

### 2.2. Is VAT Reduced in Response to Chronic Exercise or Hypocaloric Diets?

Evidence from a systematic review and meta-analysis clearly established that an increase in energy expenditure (aerobic exercise), or a decrease in energy intake (hypocaloric diet), are both associated with significant reductions in VAT, measured using gold standard methods [2]. Ross and colleagues were the first to investigate whether differences in VAT reduction were observed in response to equivalent diet- versus exercise-induced weight loss in either men [9] or women [10]. These RCTs were characterized by control of both energy expenditure (supervised exercise) and energy intake (self-reported intake every day). VAT was quantified using MRI. In these trials, participants in the exercise group did not increase or decrease their energy intake throughout the intervention, whereas those in the hypocaloric diet group did not increase physical activity levels. The primary finding from these trials revealed that when the diet- and exercise-induced weight loss was carefully matched, reductions in abdominal obesity and VAT in response to 12–14 weeks of exercise or diet were similar. Regardless of biological sex, a ~7% weight loss was associated with a ~25% reduction in VAT in response to diet or exercise^1^. These observations are consistent with the recent systematic review of Verheggen et al., wherein the results from eight studies in adults revealed that diet or exercise is associated with similar reductions in VAT when the negative energy balance and duration of intervention is matched [2]. Additionally, in response to an 18-month diet and exercise-based intervention among 278 sedentary adults with abdominal obesity, Gepner and colleagues reported that diet combined with exercise results in greater reductions in VAT than diet alone, assessed using MRI [11]. In summary, regardless of age or biological sex, a negative energy balance induced by diet or exercise is associated with marked reductions in VAT that are not materially different.

#### 2.2.1. Exercise Amount and VAT

While the evidence reviewed clearly establishes that regular exercise combined with a healthful diet is associated with a marked reduction in VAT, independent of age and biological sex, the separate effects of exercise amount and intensity on VAT are less clear. The findings from a systematic review suggest that a dose-response relationship exists between exercise amount, defined by the metabolic equivalent of task (MET-hours/week), and reductions in VAT in individuals without metabolic disorder [12]. All 16 studies included in the review measured VAT using CT or MRI.

The findings from the few randomized trials specifically designed to examine the independent contributions of exercise amount on VAT suggested that increasing exercise amount (caloric expenditure) is not positively associated with VAT reduction. Keating et al. randomized inactive, obese adults to eight weeks of either; (i) low to moderate intensity, high amount aerobic exercise (50% VO_2_ peak, 60 min); (ii) high intensity, low amount aerobic exercise (70% VO_2_ peak, 45 min); (iii) low to moderate intensity, low amount aerobic exercise (50% VO_2_ peak, 45 min); or iv) control [13]. The authors’ primary observation was that exercise conditions varying in amount (minutes) resulted in MRI-measured VAT reductions of a similar magnitude. Similarly, Slentz et al. randomized 175 sedentary, overweight adults to a control group, or for eight months, to one of three exercise groups: (i) low amount, moderate intensity, equivalent to walking 12 miles/week (19.2 km/week) at 40%–55% of VO_2_ peak; (ii) low amount, vigorous intensity, equivalent to jogging 12 miles/week at 65%–80% of VO_2_ peak; or (iii) high amount, vigorous intensity, equivalent to jogging 20 miles/week (32.0 km) [14]. Despite substantial differences in the amount (caloric expenditure) of exercise performed between groups, the primary finding was that there was no difference in CT-measured VAT reduction [12]. More recently, Cowan et al. randomized 103 previously sedentary, abdominally obese adults to one of four groups: (i) control; (ii) low amount, low-intensity exercise (180 kcal/session (women) and 300 kcal/session (men) at 50% VO_2_ peak); (iii) high-amount, low-intensity exercise (HALI; 360 kcal/session (women) and 600 kcal/session (men) at 50% VO_2_ peak); or (iv) high-amount, high-intensity exercise (HAHI; 360 kcal/session (women) and 600 kcal/session (men) at 75% VO_2_ peak) for 24 weeks [15]. Consistent with prior findings, MRI-measured VAT was reduced by 15% to 20% by comparison to controls across exercise groups, despite substantial differences in exercise-induced energy expenditure between groups. Analysis of daily self-reported diet records suggested that change in dietary intake did not differ between groups. Similarly, reduction in body weight did not differ between exercise groups. Thus, increasing exercise amount (kilocalories) without changes in energy intake for six months was not associated with greater VAT reduction in adults with abdominal obesity.

#### 2.2.2. Exercise Intensity and VAT

Whether exercise intensity is associated with VAT reduction independent of exercise amount is unclear. A systematic review that compared exercise groups that varied in exercise intensity from 14 studies suggests that exercise intensity should be at least moderate to vigorous if VAT reduction is the objective [16]. It is important to note, however, that this observation did not account for possible variation in exercise amount performed between studies. Indeed, with few exceptions, preliminary findings from randomized trials specifically designed to determine the effect of exercise intensity on VAT reduction suggest that the intensity of exercise performed is not a primary determinant of change in VAT [13,14,15,17]. Irving et al. reported that while VAT reduction was not significantly different between high intensity and low intensity exercise groups, significant within-group VAT changes by comparison to control were only observed in the high intensity group [17]. Similar to the observations for exercise amount described above, Keating et al. [13] and Slentz et al. [14] reported no effect of increasing exercise intensity on VAT reduction when measured using criterion methods. These observations were confirmed by Cowan et al. who reported that the reduction in VAT was not different independent of exercise intensity ranging from 50% to 70% of VO_2_ peak performed for 24 weeks (Figure 1) [15].

These observations regarding the effects of exercise intensity on CT-measured VAT differ from a recently completed trial wherein 220 abdominally obese Chinese adults were randomized to either a high- or moderate-intensity exercise group for 12 months [18]. The primary finding was that vigorous-moderate exercise is associated with greater reductions in VAT compared to moderate exercise. However, it is important to note that in this trial, the vigorous exercise group performed supervised exercise for the entire 12 months, whereas the moderate exercise group self-reported their exercise behaviour. Self-reported physical activity behaviour is notoriously unreliable, thus confounds interpretation [19]. Self-reported physical activity can be under- or overestimated [19].

Emerging evidence has considered whether high intensity interval training (HITT) is associated with change in VAT. HITT is characterized by brief, intermittent bursts of exercise completed at a high intensity. High intensity exercise bouts are separated by brief recovery periods of inactivity or low-intensity exercise. Although there is no consensus established for the optimal durations of exercise and rest periods for HITT protocols, active bouts may be performed from 10 s to 5 min at 70–90% of VO_2_ peak or 85%–95% of peak heart rate. Recovery periods range from 30 s to 3 min. Zhang et al. randomized 47 participants to a 12-week HIIT or moderate-intensity continuous training (MICT) intervention using a cycle ergometer three times a week [20]. Participants in the HIIT group performed cycles of 4-min high intensity bouts at 90% VO_2 max_, followed by 3 min of recovery until the targeted energy expenditure (300 kJ) was achieved. The MICT group performed continuous exercise until the matched energy expenditure target was reached (300 kJ). Similar reductions in VAT were observed regardless of exercise intervention. Winding et al. randomized 29 participants to an 11-week HIIT or MICT cycling intervention involving 10 1-min intervals (95% peak workload) with 1-min recovery or 40 min (50% peak workload), respectively [21]. Despite a lower exercise-induced energy expenditure, HIIT showed similar reductions in DXA-measured VAT compared to MICT. In both studies, participants were instructed to maintain their dietary intake throughout the intervention.

In summary, aerobic exercise combined with a balanced diet, wherein the participant does not increase energy intake, is associated with a robust reduction in VAT independent of amount or intensity. A 7% weight loss resulting from a lifestyle-based strategy consistent with current recommendations is likely to result in a 25% reduction in VAT. These observations provide treatment options for practitioners who target VAT reduction as part of a comprehensive lifestyle-based strategy designed to reduce health risk.

#### 2.2.3. Individual Variability in VAT Response to Standardized Exercise

Since all adults acquire and inherit characteristics that vary substantially, their response to a treatment designed to change a given trait will vary substantially. Although increasing exercise is a primary determinant of improvement in numerous traits (e.g., cardiorespiratory fitness) at the group level, there is a growing body of evidence that the response to regular exercise varies substantially among individuals [22]. The extent to which individual variability in VAT response to a standard dose of exercise has only recently been considered. Brennan et al. performed a study to determine the effect of exercise amount and intensity on the proportion of individuals for whom the MRI-measured VAT response was above the minimal clinically important difference, defined as a reduction in VAT greater than 0.28 kg or 9%, and whether clinically meaningful changes in waist circumference, defined as greater than 2 cm, reflect individual VAT responses that are above the minimal clinically important difference [23]. Abdominally obese men and women were randomized to (i) control (*n* = 20); (ii) low amount, low intensity (*n* = 24); (iii) high amount, low intensity (*n* = 30); or (iv) high amount, high intensity (*n* = 29) treadmill exercise for 24 weeks.

Inspection of the individual responses to standardized exercise illustrated in Figure 2 clearly document the extraordinary variability in VAT response to exercise independent of exercise amount or intensity. The observation that most adults reduce VAT regardless of exercise strategy and that clinically important reductions in VAT can be determined by corresponding reductions in waist circumference that exceed 2 cm is encouraging (Figure 2). These observations underscore the value of routinely measuring waist circumference to determine the efficacy of treatment designed to reduce VAT and that, for some adults, substantial reduction in VAT may require altering the exercise dose by increasing either the amount and/or intensity of exercise. Whether some adults are resistant to reduction in VAT regardless of exercise dose is unknown and should be the subject of future investigations.

## 3. Liver Fat (LF)

### 3.1. Measurements of LF

Liver biopsy is an established method to assess liver fat and diagnose the severity of NAFLD with precision [6]. However, the invasive nature, cost, sampling error, and morbidity associated with the procedure are acknowledged limitations [6]. CT is a technique that estimates steatosis by quantitatively measuring liver density. However, the sensitivity and specificity of CT are clinically unacceptable at mild steatosis levels [6,16]. MRI is a technique that uses quantitative and qualitative information to assess intrahepatic fat content. The heterogeneity of the magnetic field, time demanding nature, low sensitivity, and poor image quality when disturbed are limitations of this technique [6]. Ultrasound is a common technique that is used to assess LF because of the low cost, vast availability and noninvasive nature [6]. Ultrasonography is limited as a single diagnostic tool because of the lack of sufficient sensitivity [6,24,25]. Accuracy of an ultrasound decreases with increasing body mass index [26,27,28]. Magnetic resonance spectroscopy (MRS) is cited as the gold standard MR technique to quantify LF. There is a high level of accuracy and reproducibility with this method, however, it is limited in availability because of the high degree of skill and specialized software required. Similar to liver biopsies, only a portion of the liver is analyzed, and so, sampling errors may occur [26].

### 3.2. Is Liver Fat Reduced in Response to Chronic Exercise or a Hypocaloric Diet?

As with VAT, evidence from systematic reviews and meta-analyses confirm that exercise or a hypocaloric diet is associated with a reduction in LF in previously sedentary, overweight and obese men and women [3,29].

### 3.3. Exercise and Liver Fat

Evidence from observational studies have established that adults with elevated levels of LF or NAFLD are physically inactive by comparison to those without NAFLD. Kistler and colleagues performed a retrospective analysis on 813 adults with biopsy-proven NAFLD, enrolled in the Nonalcoholic Steatohepatitis Clinical Research Network (NASH CRN) [30]. Physical activity levels were derived from self-reported responses to a questionnaire. The principle finding of this cross-sectional study was that neither moderate-intensity exercise nor total exercise per week was associated with NASH. However, vigorous exercise was associated with a decreased risk of having NASH (odds ratio (OR): 0.65 (0.43–0.98)). Thus, exercise intensity may be an important consideration when prescribing exercise to reduce LF. A limitation of this study was that exercise behaviour was assessed by self-report, which is notably unreliable [19].

Based on data derived from the National Health and Nutrition Examination Survey (NHANES) 2003–2006, Gerber and colleagues evaluated the associations between levels of physical activity and NAFLD in a large subsample of adults from this database [31]. NAFLD was defined as a fatty liver index >60 in the absence of other chronic liver diseases. The fatty liver index of hepatic steatosis is calculated based on measures of triglycerides, body mass index, waist circumference, and gamma-glutamyl transpeptidase. Physical activity levels were measured objectively over seven days using accelerometry. Counter to the findings of Kistler et al. [30], Gerber and colleagues observed that adults with NAFLD (*n* = 1263), who were older and had a higher body mass index, perform less physical activity at any intensity level compared to adults without NAFLD (*n* = 1793) [31]. Thus, from initial epidemiological evidence, adults with NAFLD were physically inactive by comparison to adults without NAFLD, suggesting that exercise may be an important strategy for reducing LF. Physical activity was reported using accelerometers, which objectivity measure physical activity frequency and duration in a minimally invasive manner. However, accelerometers are limited in detecting non-ambulatory activities [19].

Evidence from controlled trials confirm the potential of exercise as a strategy to manage LF. Keating and colleagues reported that, in pooled data from six studies (156 overweight and abdominally obese adults), aerobic type exercise alone (no hypocaloric diet) ranging from two to 24 weeks at intensities between 45% and 85%, resulted in modest, yet significant reductions in LF (effect size 0.37) [29].

Keating et al. randomized 48 inactive overweight or obese adults to one of four groups varying in exercise amounts and intensities that are generally consistent with current exercise guidelines for 8 weeks [13]. LF was measured by MRS. The major finding was that LF was reduced by 1% to 3% compared to controls independent of exercise amount or intensity. The authors noted that, while the reductions in LF were modest, they were observed in association with minimal or no change in body weight.

In a much larger trial, Zhang et al. randomized 220 adults with NAFLD to 150 min per week for 12 months of either brisk walking (moderate intensity exercise) or jogging (moderate-vigorous intensity exercise) [18]. In this study, LF was assessed using MRS. All participants were asked not to change their diet during the trial. The principle finding was a decrease in LF that approximated 6% compared to controls, with no difference between exercise groups—thus, exercise intensity was not associated with LF reduction. In this trial, after controlling for the exercise-induced weight loss (3%–6%), the net changes in LF content were reduced and became nonsignificant between the exercise and control groups. Thus, the benefit of exercise was explained in large measure by the ability to induce weight loss.

Few studies have evaluated the association between cardiorespiratory fitness and LF. Pälve and colleagues found that in 463 adults, participants with fatty liver had lower cardiorespiratory fitness levels compared to participants without fatty liver (VO_2 peak_ 27.2 mL· kg^−1^·min^−1^ and 31.6 mL·kg^−1^·min^−1^, *p* < 0.0001) [32]. LF was assessed using ultrasound. This relationship remained significant when adjusted for physical activity, adiposity, smoking, alcohol consumption, serum lipids, insulin, glucose, and C-reactive protein. Participants with abdominal obesity (waist circumference > 80 cm in women and >94 cm in men) with higher fitness levels (higher than age- and sex-specific median of VO_2 Peak_) had lower prevalence of fatty liver than participants who were obese and unfit (below median), (11.7% vs. 34.8%, *p* = 0.0003). In a longitudinal study (nine months of follow up), Kantartzis et al. found that cardiorespiratory fitness (VO_2_ peak) at baseline was a strong predictor of change in MRS-measured LF in response to an exercise intervention. Independent of total fat, visceral fat, subcutaneous fat or exercise intensity, cardiorespiratory fitness was negatively associated with LF at baseline [33].

Exercise-induced reductions in intrahepatic fat are observed independent of weight change [34]. A systematic review by Hashida and colleagues showed a 20-30% relative risk reduction in LF with exercise in the absence of weight loss [35]. Sullivan and colleagues found that the effect of MRS-measured LF following an exercise intervention, according to the physical activity guidelines (150–300 min of moderate intensity exercise), were independent of weight loss [36]. In previously sedentary adults, body weight, body fat mass, and fat-free mass did not change after the intervention. However, there was a decrease (10.3% ± 4.6%) in LF in the exercise group compared with the control group (*p* = 0.04). Interestingly, obese individuals who consumed moderate to excessive amounts of alcohol underwent a 12-week exercise intervention. In this randomized trial, exercise induced reductions in both subcutaneous and visceral fat, however, there was no reduction of MRI-measured LF. This finding suggests that moderate to excessive amounts of alcohol consumption may attenuate the beneficial effects of exercise on NAFLD [37].

Qiu and colleagues conducted a meta-analysis and observed that the amount of physical activity in men was inversely associated with the risk of NAFLD in a dose-dependent manner [38]. The authors report that 500 MET-minutes/week (approximately 150 minutes/week) of physical activity was associated with an 18% risk reduction of NALFD (RR = 0.82, 95% CI 0.73–0.91). Further increases in physical activity (1000 MET-minutes/week) were associated with a 33% risk reduction in NAFLD (RR = 0.67, 95% CI 0.54–0.83). These findings are consistent with those of Li and colleagues who reported that moderate- and vigorous-intensity physical activity effectively reduced the risk of NAFLD independent of energy intake and sedentary time (>684 MET-minutes/week compared to none: OR 0.58, 95% CI 0.40 to 0.86, vigorous-intensity physical activity: >960 MET-min/week compared to none: OR 0.63, 95% CI 0.41 to 0.95) [39]. Li et al. used ultrasound measurements to assess LF. Kwak et al. also found an inverse dose-response relationship between the amount of self-reported physical activity and the prevalence of NAFLD, independent of insulin resistance and VAT in 3718 adult men and women [40]. However, self-reported physical activity behaviour is notoriously unreliable, thus confounding interpretation [19].

Emerging evidence has also considered whether HITT is associated with change inLF. Hallworth and colleagues randomized participants to 12 weeks of HITT on a cycle ergometer three times a week. Participants were instructed to perform 5 intervals of cycling at 16–17 on the Borg scale for 3 min followed by 3 min of recovery [39]. Abdelbasset et al. randomized participants to an eight-week cycling protocol. Participants completed three sets of 4-min cycling at 80 to 85% of VO_2_ max with 2-min recovery, three days a week [40]. In both studies, participants were instructed to make no modifications to their diets. Several authors have observed reductions in LF after HIIT compared to controls [41,42]. The change in LF was measured using MRI and/or ultrasonography. When HITT and moderate intensity exercise were matched for energy expenditure, there were no differences in LF reduction [41,42,43].

### 3.4. Hypocaloric Diet and LF

It is well established that diet-induced weight loss is a cornerstone of treatment for persons with NAFLD. In 2003, Tiikkainen and colleagues reported that LF, as measured by MRS, was reduced by about 39% in premenopausal women consequent to an 8% weight loss achieved within three to six months [43]. Of note, the authors reported that the reduction in LF was directly related to baseline levels. Larson-Meyer et al. reported results from the CALERIE study wherein a loss of body weight approximating 10% via hypocaloric diet or diet and exercise combined, was associated with a 29% to 40% reduction in MRS-measured LF in overweight adults. As with exercise-induced weight loss, diet-induced weight loss is associated with a marked reduction in MRI-measured LF [11]. Very low-calorie diets also resulted in significant reductions of LF, however, very low-calorie diets cannot be maintained long term [44,45,46].

Thoma and colleagues performed a systematic review of the literature to evaluate the effects of various lifestyle-based interventions on LF [3]. For diet only interventions, the authors uncovered 10 studies wherein 11 groups totaling 322 participants were prescribed a hypocaloric diet varying in composition and negative energy balance. Only two studies included control groups. The duration of the diet interventions ranged from one to six months. The mean weight loss across the studies ranged from 4% to 14% and the relative reduction in LF, as measured by MRS, ranged from 42% to 81%. No clear relationship between diet composition and LF reduction was reported. However, consistent with the evidence cited above, the reduction in LF was strongly associated with weight loss.

### 3.5. Exercise and Hypocaloric Diet Combined

Lifestyle interventions that include a hypocaloric diet and an exercise intervention have been considered in several systematic reviews. The reviews investigate if a combination of diet and exercise is more effective than exercise or diet alone to decrease LF [3,29,47,48,49]. Keating and colleagues did not find a significant pooled effect of combined interventions of diet and exercise; however, the authors discuss being limited by a low sample size and statistical power [29]. Thoma et al. reported on seven studies that conducted lifestyle modification interventions to decrease energy intake and increase physical activity or exercise over 3–12 months [3].The authors concluded that lifestyle modifications consistently resulted in LF reduction. Similarly, other systematic reviews conducted by Hens and colleagues, as well as Whitsett and colleagues, found that lifestyle modification can reduce LF [48,49]. Golabi et al. found that exercise alone and a combination of diet and exercise both significantly decrease LF [47].

## 4. Anthropometric Markers of VAT and LF

Criterion measurement of both VAT and LF requires sophisticated radiological techniques (e.g., magnetic resonance imaging or computed tomography) that are not readily accessible in most clinical settings. LF can also be determined using biopsy methods, but this too is invasive and presents a burden to the patient. Accordingly, the validity of simple anthropometric methods has been considered as an alternative, pragmatic option. There is general agreement that waist circumference represents the single best anthropometric marker of VAT [1,50]. However, due in large measure to the inter-individual variability between the quantity of abdominal subcutaneous and VAT [51], the variance explained in VAT change by waist circumference is modest, ranging from 25% to 75%^1^. At present, there is no universally accepted method for the measurement of waist circumference. The two most often used protocols are at the level of the iliac crest and the mid-point between the iliac crest and last rib [52]. Whether the associations between these two WC methods provide different values for VAT across sex, age, and ethnicity is unclear.

In 2006, Bedogni et al. derived a simple index (Fatty Liver Index) of hepatic steatosis that was calculated based on the measures of triglycerides, body mass index, waist circumference and gamma-glutamyl transpeptidase (GGT) [53]. Fatty liver index has a very good discriminative ability to predict fatty liver in both Asian and Western adult populations [54,55]. Motamed et al. recently validated the discriminative ability of the Fatty Liver Index to predict fatty liver in a sample of 5052 adults [56]. Fatty liver or NAFLD was determined by ultrasound. Interestingly, although the Fatty Liver Index showed good discriminative ability for the diagnosis of NAFLD (AUC = 0.87 (95%CI: 0.85–0.88), there was no significant difference in the discriminative ability determined by waist circumference alone (AUC = 0.85, 95%CI: 0.84–0.86). Thus, on a population basis, waist circumference alone may be a useful measure of both VAT and LF. However, the utility of either waist circumference or Fatty Liver Index to predict VAT and LF, respectively, on an individual basis is unknown.

## 5. Summary

It is now firmly established that both regular aerobic exercise and the consumption of a hypocaloric diet are associated with a substantial reduction in VAT and LF independent of age, biological sex, or ethnicity. Whereas exercise is associated with VAT reduction with or without weight loss, reductions in LF are positively associated with weight, independent of the strategy to induce weight loss. Despite decades of research into the associations between lifestyle-based interventions and VAT reduction, the threshold of VAT reduction that is required for health benefit remains unclear. Similarly, while it is clear that weight loss is associated with LF reduction, optimal levels of weight loss for attenuating LF are unclear. Nevertheless, it is extremely encouraging that regular exercise (4–6 months) consistent with consensus recommendations (30–60 min per day at moderate-to-vigorous intensity; e.g., brisk walking or jogging) combined with a balanced, healthful diet is associated with substantial reductions in VAT (15%–20%). Similarly, weight loss of 5% to 10% can be achieved by 4–6 months with reasonable reductions in caloric intake with or without exercise. Thus, practitioners in health care settings have options when counselling adults regarding the utility of lifestyle-based interventions designed to reduce both VAT and LF. The objective of this review was assessing strategies for reducing VAT and LF. The focus was on a negative energy balance induced aerobic exercise and a decrease in energy intake. Future research is required to determine if some adults are resistant to reductions in VAT and LF regardless of exercise dose.

Direct measurement of VAT or LF in most health care settings is not feasible. Waist circumference remains the single best anthropometric marker of change in VAT. While precise measurement of change in VAT using waist circumference on an individual basis is unlikely, it is extremely likely that reductions in waist circumference (e.g., greater than 2 cm) are associated with a corresponding reduction in VAT. While initial results suggest that waist circumference may be as useful as the LF Index to follow change in LF, additional evidence from large scale studies in diverse populations is required. Nevertheless, these observations are encouraging and reinforce the importance of measuring waist circumference in all health care settings.

## Figures and Tables

**Figure 1 nutrients-12-00891-f001:**
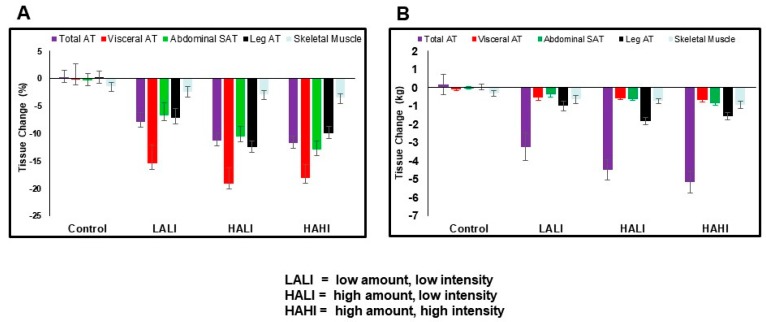
Illustration of relative (%, Panel **A**) and absolute (kg, Panel **B**) response of adipose and skeletal muscle tissues to variations in exercise dose. Taken from Reference [14]. Used with permission from Obesity. Absolute (left) and relative (right) changes in AT and skeletal muscle depots. With the exception of skeletal muscle, all AT depot changes (absolute and relative) were significantly different from control (*p* < 0.008; both panels). Relative change in VAT was greater than all other AT and skeletal muscle depots (*p* < 0.01; right panel). AT, adipose tissue; HAHI, high amount, high intensity; HALI, high amount, low intensity; LALI, low amount, low intensity.

**Figure 2 nutrients-12-00891-f002:**
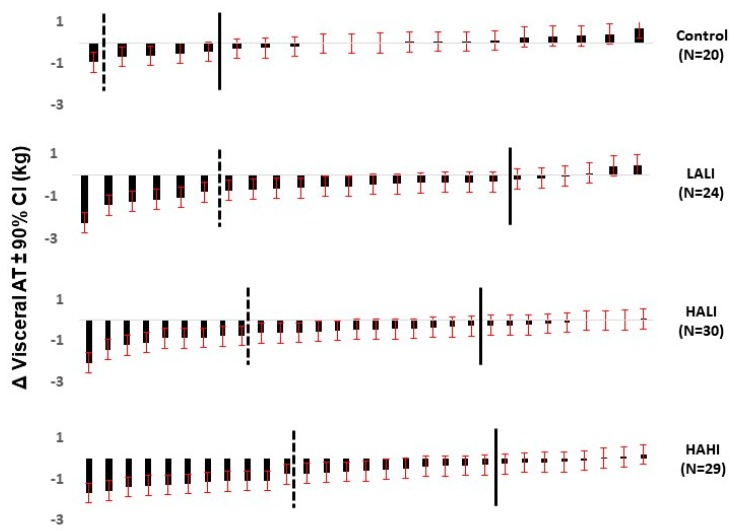
Variability of visceral adipose tissue (VAT) response to standardized exercise. Taken from Reference [19]. Used with permission from MSSE. Distribution of individual responses for change in VAT (kg) ± 90% CI in the control and intervention groups. The solid black line distinguishes participants whose observed response exceeds the MCID (e.g., those participants to the left of the solid line reduced visceral AT by < 0.28 kg). The dashed black line distinguishes participants whose observed response ± 90% CI exceeds the MCID (e.g., those participants to the left of the dashed line have CI where the top range is < 0.28 kg). LALI (low amount (~30 min), low intensity (50% VO_2_ peak)), HALI (high amount (~60 min), low intensity (50% VO_2_ peak)), HAHI (High amount (~40 min), high intensity (75% VO_2_ peak)).

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
