# Peer review of "Negative Energy Balance Induced by Exercise or Diet: Effects on Visceral Adipose Tissue and Liver Fat"

_nutrients, 2020, doi:10.3390/nu12040891_

Round 1

Reviewer 1 Report

Peer review

This is an interesting review of an important area namely the effect of energy deficit on stores of visceral and intrahepatic fat. It is well written and organised overall. Below are some recommendations that will factor in my recommendation for publication, and some that are simply suggestions to enhance the usefulness of the review should the 

Recommendations:

Despite this being clearly described as a narrative review, the review would benefit from some further critical commentary on the relevant methods employed in the studies cited. This is done for self-reported physical activity, but less so for dietary intake estimates, and importantly measures of visceral adipose tissue for which even volume vs. cross-sectional area using the same technology can make a difference (e.g. see Shuster et al., 2012). The limitations of the studies sighted should be made clear to a reader who may not themselves be sufficiently familiar with assessment methods or study design/methodology limitations.

On the same note, the method(s) used in any given study should be clear so that the reader knows when studies have used the same/similar methods and when they have not, and therefore what the limitations of the measurement method are.

The reader would benefit from some recommendations about what future research in this area should look like to fill in the gaps identified, tracking energy balance with doubly labelled water as both energy intake and most energy expenditure estimates are inaccurate, using accelerometry to track activity.

Line 25: It would help the reader to have the association between CVD and VAT quantified in some way, or at least have the cardiometabolic risk factors explicitly named.

Line 43-44: “Unlike VAT, the utility of exercise or diet to reduce liver fat 44 appears to be associated with weight loss.” This statement does not reflect exercise studies that have shown meaningful intrahepatic fat reductions with no or minimal weight change, e.g. Hallsworth et al (2011 and 2015), Houghton et al (2017) and ...

Line 53: The word “exercise” would be better replaced with “physical activity” as this is the broader term.

Line 56-57: The sentence implies that self-report of dietary intake represents strict control, this is not consistent with the known frequency of mis-reporting of dietary intake. Likewise, while exercise may have been supervised, this does not guarantee that change in energy expenditure is known as a common finding is that an increase in energy expenditure during exercise is accompanied by a decrease in non-exercise physical activity. A metabolic ward study would represent ‘strict control’.

Line 82: The sentence refers to exercise “amount”, and later on it appears that this may refer to METs or time/duration, but despite the concept of “amount” or even “dose” being relevant to physical activity, there is no way to accurately quantify this as it is made up of frequency, intensity, time (duration) and type all of which are defined by the American College of Sports Medicine. It would therefore be better to simply say “… exercise frequency, intensity, time, and type, and then subsequently refer to the exercise “time”.

Line 85: What is the quantified relationship/correlation?

Line 88-91: What was the frequency (days/week) of the exercise? There appears to be only a 25% difference in time for “low” and “high” amount, so frequency is highly relevant to work out the actual difference in total time/week.

Line 99-107: In relation to Cowan et al (2018), were dietary changes reported and factored in in any way? Was weight reduction similar between groups, if not, was this factored considered?

Line 140: The section is unclear – does it refer to an increase in physical activity related energy expenditure without an increase in energy intake thereby creating a small energy deficit, or a “eucaloric diet” in which energy intake and energy expenditure are equal? If the latter, the term “eucaloric diet” should replace “balanced diet”, as the latter may be interpreted to mean something other than what is described.

Line 141: It would be preferable to quantify, perhaps as a range of averages, the reduction in VAT as opposed to simply saying it is “robust”; this would also be in line with the next part of the paragraph.

Line 151-152: The sentence “The extent to which …” is incomplete

Line 163: Use VAT rather than visceral AT, to be consistent with the rest of the text.

Line 183-260: Some key papers looking at intrahepatic fat modification by exercise are missing , i.e. Hallsworth et al (2011 and 2015), Houghton et al (2017) and , as is the paper by Cassidy et al (2016), which suggests modification of liver fat may be mediated by degree of metabolic derangement.

Line 193: Although the comment about self-report is very consistent with the literature, a reference would be helpful to some readers. Also, the common direction (under vs. over-reporting) should be mentioned.

Line 204: The paper by Hallsworth et al (20.  Also using accelerometry is relevant here.

Line 219: Because liver fat itself is reported as a percentage, it is important to be very clear about the size of the reduction, 6% could be taken to mean 12% down to 6% or 10% down to 9.4%. This is easily solved by quoting actual means and variances, which also provides more information about the sample characteristics of a study.

Line 261 to 279: A number of studies assessing very low calorie diets and low carbohydrate diets and changes in liver fat appear to be missing here: e.g. Lewis et al (2006), Lim et al (2011),  Browning et al (2006), and Gu et al (2013). The latter relate to an earlier recommendation to consider dietary composition in the discussion. Further, these papers would allow some discussion of the speed of change, something both clinicians and patients are likely to be interested in.

Suggestions

The focus of the review is clearly on energy, however mention is made of ‘diet quality’. Diet quality scores assume a single broad category of healthy diet exists, this is contrary to the totality of the evidence, which shows us that several dietary approaches can promote or maintain health. It would be more helpful to have diet composition receive some attention.

A general comment that may inform further critique – extreme caution is warranted when interpreting meta-analyses that try to pool data from studies using fundamentally different interventions, e.g. diet in one and exercise in another, different populations, and outcome measures assessed using different techniques, unless these techniques have been thoroughly tested head-to-head and found to be consistent or predictably inconsistent. Although meta-anlyses are a powerful tool for pooling data when interventions and study populations are very similar, e.g. drug trials with the same medication, they are often less well suited to exercise in which dose cannot be well defined as it is a composite of frequency, intensity, time, and type, or diet which likewise has a myriad of components beyond total energy content.

Contrasting of estimated energy deficit and length of time – is rapid weight reduction more/less/similarly effective at reducing VAT?

Figure 2: the lower bound of the error bars is hard to see. Perhaps a suitable app could be used to change the colour.

Line 177-178: this comment deserves to be repeated in the summary as it is an important point of uncertainty.

Line 294: It could also be noted that biopsy estimated liver fat is subject to inaccuracy due to heterogenous distribution of liver fat.

Line 310: It may be worth noting that ultrasound is a relatively inaccurate way of assessing liver fat, and is most reliable at higher 10-12+%, and so represents a less than ideal way of validating the Fatty Liver Index.

Author Response

Reviewer # 1

This is an interesting review of an important area namely the effect of energy deficit on stores of visceral and intrahepatic fat. It is well written and organised overall. Below are some recommendations that will factor in my recommendation for publication, and some that are simply suggestions to enhance the usefulness of the review should the….. 

Comment:  Despite this being clearly described as a narrative review, the review would benefit from some further critical commentary on the relevant methods employed in the studies cited. This is done for self-reported physical activity, but less so for dietary intake estimates, and importantly measures of visceral adipose tissue for which even volume vs. cross-sectional area using the same technology can make a difference (e.g. see Shuster et al., 2012).

Response:  We appreciate the reviewer’s comments regarding the addition of insight related to the methods employed to measure VAT and LF.  We have revised the manuscript accordingly (please see page 2 Measurement of VAT and LF).  For the reviewer’s information, we note that within the reference suggested (Shuster and colleagues (2012)), the references that address associations between changes in VAT determined using a single image report that they are not materially different from the corresponding reduction in VAT volume derived using a multiple-image protocol.  In fact, the references cited by Shuster and colleagues originate from our research group. 

Comment: On the same note, the method(s) used in any given study should be clear so that the reader knows when studies have used the same/similar methods and when they have not, and therefore what the limitations of the measurement method are.

Response:  Please note that we now include reference to the methods used to assess VAT and LF for the studies cited.  Details, albeit brief, for all methods cited is now given (see page 2, lines 59-65 and page 7, lines 211-226).   

Comment: The reader would benefit from some recommendations about what future research in this area should look like to fill in the gaps identified, tracking energy balance with doubly labelled water as both energy intake and most energy expenditure estimates are inaccurate, using accelerometry to track activity.

Comment: Line 25: It would help the reader to have the association between CVD and VAT quantified in some way, or at least have the cardiometabolic risk factors explicitly named.

Response:  Although we appreciate the basis for this request, the sheer volume of evidence that we would need to consider to fully respond as requested is beyond the scope of this review.  However, the reference we provide alerts the reader to a recently published consensus statement from an international consortium of experts wherein the reader can review the related evidence.

Comment:  Line 43-44: “Unlike VAT, the utility of exercise or diet to reduce liver fat appears to be explained in large measure by weight loss.” This statement does not reflect exercise studies that have shown meaningful intrahepatic fat reductions with no or minimal weight change, e.g. Hallsworth et al (2011 and 2015), Houghton et al (2017) and ...

Response: Agreed, this sentence has been removed to ensure consistency.

Comment:  Line 53: The word “exercise” would be better replaced with “physical activity” as this is the broader term.

Response:  The reviewer makes a good point that it is important to distinguish between exercise and physical activity. However, the interventions within the cited sources specifically focus on exercise, rather than the broader term – physical activity. Although it would be more appropriate to use physical activity when referring to observational studies that include a broad range of energy-expending activities (i.e. activities of daily living, exercise, active transportation, etc.), the RCTs included in our paper specifically examine “exercise”.

Comment: Line 56-57: The sentence implies that self-report of dietary intake represents strict control, this is not consistent with the known frequency of mis-reporting of dietary intake. Likewise, while exercise may have been supervised, this does not guarantee that change in energy expenditure is known as a common finding is that an increase in energy expenditure during exercise is accompanied by a decrease in non-exercise physical activity. A metabolic ward study would represent ‘strict control’.

Response: We appreciate the reviewers comment and have removed the word ‘strict’ from the sentence in question (see line 77).

Comment: Likewise, while exercise may have been supervised, this does not guarantee that change in energy expenditure is known as a common finding is that an increase in energy expenditure during exercise is accompanied by a decrease in non-exercise physical activity. A metabolic ward study would represent ‘strict control’.

Response: We note that the reviewer did not provide evidence to support the argument that a change in energy expenditure is accompanied by a decrease in non-exercise physical activity. To the contrary, a systematic review (see Washburn and colleagues (citation below)) examining thirty-one studies concludes that there is limited evidence to support the hypothesis that prescribed exercise training results in decreased non-exercise physical activity energy expenditure.  

Washburn, R. A., Lambourne, K., Szabo, A. N., Herrmann, S. D., Honas, J. J., & Donnelly, J. E. (2014). Does increased prescribed exercise alter non‐exercise physical activity/energy expenditure in healthy adults? A systematic review. Clinical obesity4(1), 1-20.

Comment:  Line 82: The sentence refers to exercise “amount”, and later on it appears that this may refer to METs or time/duration, but despite the concept of “amount” or even “dose” being relevant to physical activity, there is no way to accurately quantify this as it is made up of frequency, intensity, time (duration) and type all of which are defined by the American College of Sports Medicine. It would therefore be better to simply say “… exercise frequency, intensity, time, and type, and then subsequently refer to the exercise “time”.

Response: Perhaps we miss the authors point. Frequency per se is not relevant when quantifying in MET/hours per week and the modality of exercise addressed in our review is restricted to aerobic activity. The author is also well aware that it is not possible to determine exercise intensity when exercise is expressed in MET/hours per week.  Perhaps the more important point made in this section of the review is that when exercise amount and intensity are clearly identified within the design of a randomized controlled trial, exercise amount (eg expressed in exercise-induce energy expenditure (kilocalories) is not related to VAT reduction.  As we note in this section, a rationale that would explain this observation remains to be determined.  

Comment: Line 85: What is the quantified relationship/correlation?

Response: The relationship has been quantified. Please refer to line 108.

Comment:  Line 88-91: What was the frequency (days/week) of the exercise? There appears to be only a 25% difference in time for “low” and “high” amount, so frequency is highly relevant to work out the actual difference in total time/week.

Response:  We agree that frequency is highly relevant in determining the exercise amount prescribed per week. However, the primary conclusion drawn within the referenced study (Keating et al., reference 12) is that the observed reductions in VAT occurred independent of exercise amount. Thus, we did not cite the variation in exercise frequency.

Comment: Line 99-107: In relation to Cowan et al (2018), were dietary changes reported and factored in in any way? Was weight reduction similar between groups, if not, was this factored considered?

Response:  Thank you.  We have revised the manuscript to indicate that both dietary intake and body weight change did not differ between exercise groups. See lines 134-135.

Comment:  Line 140: The section is unclear – does it refer to an increase in physical activity related energy expenditure without an increase in energy intake thereby creating a small energy deficit, or a “eucaloric diet” in which energy intake and energy expenditure are equal? If the latter, the term “eucaloric diet” should replace “balanced diet”, as the latter may be interpreted to mean something other than what is described.

Response:  As indicated within the original text, the observations are consequent to a participation in exercise without an increase in energy intake. Thus, the negative energy balance is induced by exercise.  In other words, no compensation in energy intake was prescribed.  We believe that we state this clearly by stating that: “exercise combined with a balanced diet wherein the participant does not increase energy intake.” (refer to line 171)

Comment:  Line 141: It would be preferable to quantify, perhaps as a range of averages, the reduction in VAT as opposed to simply saying it is “robust”; this would also be in line with the next part of the paragraph.

Response:  We refer the reviewer to the text immediately below that in question wherein we report a 15-25% reduction in VAT resulting from a lifestyle-based intervention (please see lines 173-174).

Comment: Line 151-152: The sentence “The extent to which …” is incomplete

Response:  This sentence has been revised. Please see line 182-183.

Comment:  Line 163: Use VAT rather than visceral AT, to be consistent with the rest of the text.

Response: We agree. The text has been revised as suggested. Please see line 195.

Comment: Line 183-260: Some key papers looking at intrahepatic fat modification by exercise are missing , i.e. Hallsworth et al (2011 and 2015), Houghton et al (2017) and , as is the paper by Cassidy et al (2016), which suggests modification of liver fat may be mediated by degree of metabolic derangement.

Response:  We appreciate the reviewers’ recommendation.  In our original manuscript we did not state that we focused on aerobic-type exercise alone.  We have rectified this omission, please see lines 49-50. Accordingly, the Hallsworth 2011 and Houghton 2017 study are not included as the authors considered the association between resistance exercise and LF.  However, the reviewer does make the point that we had not previously considered high intensity interval training. We have revised the manuscript to include findings from high intensity interval exercise interventions have been included, please see lines 348-353.

Comment: Line 193: Although the comment about self-report is very consistent with the literature, a reference would be helpful to some readers. Also, the common direction (under vs. over-reporting) should be mentioned.

Response: Agreed, an appropriate reference is now given. Please see lines 16-187.

Comment: Line 204: The paper by Hallsworth et al (20.  Also using accelerometry is relevant here.

Response: Agree. Please see revisions, lines 242-256.

Comment:  Line 219: Because liver fat itself is reported as a percentage, it is important to be very clear about the size of the reduction, 6% could be taken to mean 12% down to 6% or 10% down to 9.4%. This is easily solved by quoting actual means and variances, which also provides more information about the sample characteristics of a study.

Response:  If we understand correctly, the reviewers comment does not apply to the observation cited wherein the 6% figure does not refer to within treatment group reduction but rather, the difference between the treatment and control group. 

Comment: Line 261 to 279: A number of studies assessing very low calorie diets and low carbohydrate diets and changes in liver fat appear to be missing here: e.g. Lewis et al (2006), Lim et al (2011), Browning et al (2006), and Gu et al (2013). The latter relate to an earlier recommendation to consider dietary composition in the discussion. Further, these papers would allow some discussion of the speed of change, something both clinicians and patients are likely to be interested in.

Response:  As we state within the introduction, the primary objective of our review was to consider diet-induced negative energy balance (eg diet quantity) not diet composition. However, as suggested by the reviewer, we now report findings from very low-calorie diet studies have been included, please see changes on line 49-50 and 366-367.

Suggestions

The focus of the review is clearly on energy, however mention is made of ‘diet quality’. Diet quality scores assume a single broad category of healthy diet exists, this is contrary to the totality of the evidence, which shows us that several dietary approaches can promote or maintain health. It would be more helpful to have diet composition receive some attention.

Response:  In the original manuscript we referred to diet composition on a single occasion (originally reference 10).  To be consistent with our primary objective to consider as stated within the introduction, that is, to restrict our review to studies that consider diet-induced negative energy balance, we now omit reference to this study from our review. 

A general comment that may inform further critique – extreme caution is warranted when interpreting meta-analyses that try to pool data from studies using fundamentally different interventions, e.g. diet in one and exercise in another, different populations, and outcome measures assessed using different techniques, unless these techniques have been thoroughly tested head-to-head and found to be consistent or predictably inconsistent. Although meta-analyses are a powerful tool for pooling data when interventions and study populations are very similar, e.g. drug trials with the same medication, they are often less well suited to exercise in which dose cannot be well defined as it is a composite of frequency, intensity, time, and type, or diet which likewise has a myriad of components beyond total energy content.

Response: We understand the reviewers observation.  Unless we missed it, we do not sense that a specific recommendation for revision was suggested.

Comment:  Contrasting of estimated energy deficit and length of time – is rapid weight reduction more/less/similarly effective at reducing VAT?

Response:  Our sense is that in general, VAT reduction is associated with diet- or exercise-induced negative energy balance in a dose-response manner.  While there are few dose-response trials (RTCs) in the current literature, our guess is that the magnitude of the negative energy balance is the primary determinant of VAT reduction.  The larger the negative energy balance, the greater the reduction. 

Comment:  Figure 2: the lower bound of the error bars is hard to see. Perhaps a suitable app could be used to change the colour.

Response:  When enlarged the error bars seem clear.  Perhaps the Editor will have suggestions for revision at the ‘proof’ stage?

Comment:  Line 177-178: this comment deserves to be repeated in the summary as it is an important point of uncertainty.

Response: Thank You.  Revised accordingly. Please refer to page 11, line 395-396.

Comment:  Line 294: It could also be noted that biopsy estimated liver fat is subject to inaccuracy due to heterogenous distribution of liver fat.

Response: Agree.  We have revised the manuscript where appropriate to indicate the methods used to determine VAT and LF.

Comment:  Line 310: It may be worth noting that ultrasound is a relatively inaccurate way of assessing liver fat, and is most reliable at higher 10-12+%, and so represents a less than ideal way of validating the Fatty Liver Index.

Response:  Agree. Please see revised manuscript, lines 242-257.

Reviewer 2 Report

Outstanding manuscript.

Minor comments:

First paragraph: the use of the word "diet" is a bit confusing at this stage of the paper.  Maybe introduce the term hypocaloric diet here and provide more detail on the concept of "minimal or no weight loss".

Line 99: looks like a citation error after vat reduction

Line 143: Not consistent with line 62-63

Author Response

Comment: First paragraph: the use of the word "diet" is a bit confusing at this stage of the paper.  Maybe introduce the term hypocaloric diet here and provide more detail on the concept of "minimal or no weight loss".

Response:  Agreed. The term hypocaloric has been added to clarify the type of diet we are referring to. See lines 31 and 34.

Comment: Line 99: looks like a citation error after vat reduction

Response: This citation has been corrected. Please see line 126.

Comment: Line 143: Not consistent with line 62-63

Response: We appreciate the reviewer’s observation. Revisions have been made. Please see lines 83-84 and 173-174.

Round 2

Reviewer 1 Report

Thank you for making the changes that now provide the reader with some more context. The point about there be no convincing evidence the changes in NEAT and NEPA is well taken, as this is also the conclusion of the most recent systematic review.